# Experimental test of photonic entanglement in accelerated reference frames

Matthias Fink[1], Ana Rodriguez-Aramendia[1], Johannes Handsteiner[1], Abdul Ziarkash[1], Fabian Steinlechner[1], Thomas Scheidl[1], Ivette Fuentes[2], Jacques Pienaar[1,2,†], Timothy C. Ralph[3] & Rupert Ursin[1,4]

The unification of the theory of relativity and quantum mechanics is a long-standing challenge in contemporary physics. Experimental techniques in quantum optics have only recently reached the maturity required for the investigation of quantum systems under the influence of non-inertial motion, such as being held at rest in gravitational fields, or subjected to uniform accelerations. Here, we report on experiments in which a genuine quantum state of an entangled photon pair is exposed to a series of different accelerations. We measure an entanglement witness for g-values ranging from 30 mg to up to 30 g—under free-fall as well on a spinning centrifuge—and have thus derived an upper bound on the effects of uniform acceleration on photonic entanglement.

[1] Institute for Quantum Optics and Quantum Information—Vienna (IQOQI), Austrian Academy of Sciences, Boltzmanngasse 3, Vienna A-1090, Austria. [2] Faculty of Physics, University of Vienna, Boltzmanngasse 5, Vienna A-1090, Austria. [3] Centre for Quantum Computation & Communication Technology, School of Mathematics and Physics, University of Queensland, Brisbane, Queensland 4072, Australia. [4] Vienna Center for Quantum Science and Technology (VCQ), Vienna, Austria. † Present address: International Institute of Physics (IIP-UFRN), Avenida Odilon Gomes de Lima 1722, 59078-400 Natal, Brazil. Correspondence and requests for materials should be addressed to M.F. (email: Matthias.Fink@oeaw.ac.at) or to R.U. (email: Rupert.Ursin@oeaw.ac.at).

Einstein's relativity theory and quantum theory are two pillars of modern physics that have each been shown to work well in their respective regimes. All currently observed physical effects fall well within one regime or the other, and can be described by the respective theory to very high precision. Without the scientific insights from these two theories, a modern information society would not be imaginable, taking the global positioning system or semiconductor technology as two prominent examples.

The unification of Einstein's theory of relativity and quantum mechanics is a long-standing problem in contemporary physics. As long as these descriptions of nature remain confined to their own scope of application, they cannot contribute to a unified theory that captures physics at the boundary between these specialized regimes. In natural science, where theory becomes uncertain, new experimental data are needed. It is not unusual for the first steps into an uncharted experimental regime to yield unexpected results, as occurred in the historical black-body radiation[1] and Michelson–Morley[2] experiments.

One phenomenon whose limits have not been fully tested in all regimes is quantum entanglement of spatially separated quantum systems. Entanglement can spread quantum superpositions over macroscopic distances and has been experimentally found to exist over distances as large as 144 km (ref. 3). The non-local character of these superpositions then seems in conflict with the local nature of relativity. This raises the question of whether entanglement persists, when we consider non-inertial reference frames, such as those experienced by accelerated systems or systems in gravitational fields, as the space-time metric can vary with position in these cases. It is likely that the resolution of such questions will be a key piece in the puzzle leading to a full theory of quantum gravity. Hence, quantum entangled systems subjected to high- and low-accelerations is one regime, where new physical phenomena might potentially arise. It is known that experimental investigations involving simulated hyper- or milligravity can cause unexpected changes to physical phenomena[4]. Exposing physical systems to such extreme conditions can aid in the understanding of that system, and lead to a deeper understanding of the physical processes themselves.

It has been suggested that motion and gravity can both have observable effects on quantum entanglement[5–8]. However, experimental tests of quantum phenomena in these regimes is still sparse[9]. To date, first experiments have been performed using only single quanta, e.g., in the pioneering work of Colella et al.[10,11]. Apart from this, there has been no systematic experimental investigation of the effect of non-inertial motion or gravity on quantum entanglement.

Just recently a remarkable one-shot experiment was carried out where a correlated photon source was exposed to ultra high acceleration during an explosion (when the rocket exploded during the launch phase of the mission), but not during it's operation[12]. In a subsequent launch this project ultimately succeeded in placing a correlated photon state on a cubesat into space[13]. Those Experiments demonstrate very high stability of optical alignment under harsh mechanical and thermal conditions but do not directly relate to properties of

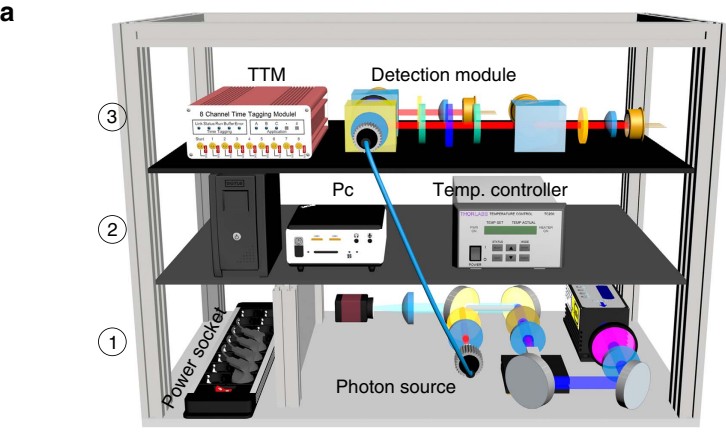

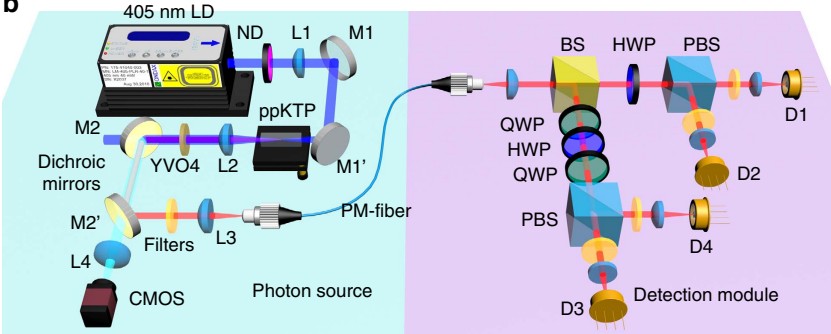

**Figure 1 | Sketch of the source crate. (b)** A laser diode pumps a ppKTP crystal heated by an oven, generating photon pairs collected in a polarization maintaining single-mode fibre. An additional $Nd:YVO_4$ crystal is used to fine-tune the walk-off compensation. A BS creates an equal superposition for reflection and transmission for each individual photon, which leads to an polarization-entangled state in postselection. Using various half- and quarter-wave plates (HWP, QWP) and PBS, the polarization correlations can be analysed in different measurement bases and measured in detectors (D1,D2,D3 and D4). (**a**) Source (level 1), electronics (level 2), as well as polarization analysis and detection module (level 3) are placed at different levels inside the crate and thus exhibit different maximal g-values.

entanglement under these conditions. However, with the onset of new technology, techniques now exist to transgress experimental barriers and begin to test quantum physics in non-inertial systems.

Following these lines we report on a series of experiments that test quantum theory in non-inertial reference frames by exposing a provably quantum state[14] of polarization-entangled photon pairs to various uniform accelerations. The acceleration is imposed by its motion, in free-fall, as well as on a centrifuge. In a first series of experiments, we dropped a crate containing an entangled photon source and two single-photon polarization detection units from 12 m in a drop tower to realize a milli-g environment, where we reached 30 mg. In the second series of experiment, the crate was mounted on an arm of a rotational centrifuge and accelerated to, as high as 30 g at a maximum angular speed of 9.9 rad s$^{-1}$.

Our results show that quantum entanglement is unaffected by non-inertial motion to within the resolution of our test-system. This represents the first experimental effort exposing a genuine quantum system to milli-g and hyper-g and extends the experimental regime in which quantum effects can be said to exist in harmony with relativity. Future experiments could be designed to investigate if the entangled state is transformed when the system undergoes non-uniform acceleration or much higher accelerations or massive gravitational fields.

## Results

**Entangled photon system.** We developed an ultra-stable entangled photon system, comprising a source of polarization-entangled photons and detection setup. The optical setup and the electronic control equipment were mounted in a rigid three-level crate (Fig. 1a), which could withstand the high g-forces during the deceleration and acceleration phases in our intended drop-tower and centrifuge experiments. The integrated crate design also facilitated flexible plug-and-play installation at the drop-tower and centrifuge support structures.

The entangled photon source (Fig. 1) is based on spontaneous parametric down conversion (SPDC) in a degenerate collinear type-II quasi-phase matching configuration and postselection on a beam splitter (BS)[15]. A continuous-wave pump laser at 405 nm is focused loosely into a periodically poled potassium titanyl phosphate (ppKTP) crystal and creates pairs of signal and idler photons with horizontal (H) and vertical (V) polarization, respectively. The temperature of the nonlinear crystal is stabilized at 39.6 °C ± 0.1 °C for wavelength-degenerate quasi-phase matching at 810 nm. To monitor possible misalignment of the beam paths during acceleration, a fraction of the pump beam was imaged onto a CMOS camera that was placed behind one of the dichroic mirrors. The photon pairs were separated from the pump light using two dichoric mirrors and an interference filter, and coupled into a sole polarization maintaining single-mode optical fibre (PMSMF). The polarization axis of the PMSMF was carefully aligned to coincide with the optical axes of the nonlinear crystal. The length of the PMSMF and an additional neodymium-doped yttrium orthovanadate (Nd:YVO$_4$) crystal were chosen, such that the average longitudinal walk-off between the signal and idler photons due to the birefringence of the ppKTP crystal, was compensated. This ensures, that no information about the polarization of a photon can be obtained from the timing of the SPDC photons. The PMSMF also had the benefit of ensuring polarization stability even at high accelerations during an experimental run. The PMSMF guides the photon pairs to a non-polarizing BS, which creates an equal superposition of reflection (mode 1) and transmission (mode 2). When post-selecting events where one photon is detected in each of these two modes, one thus obtains the maximally polarization-entangled state:

$$|\Psi\rangle = 1/\sqrt{2}\left(|H_1V_2\rangle + e^{-i\phi}|V_1H_2\rangle\right) \tag{1}$$

where H (V) denote the horizontal (vertical) polarization states relative to the baseplate of the source. Note that the orientation of the baseplate changes relative to the laboratory frame during the experiment (Fig. 2b).

A combination of a quarter-, half- and quarter-wave plate is used to compensate polarization dependent phase shifts caused by the reflection at the BS and allows setting the phase $\phi$. For the presented experiments the quarter-wave plate was set such that $\phi = \pi$ resulting in the rotationally invariant two-photon state $|\Psi^-\rangle = 1/\sqrt{2}(|H_1V_2\rangle - |V_1H_2\rangle)$. An additional motorized half-

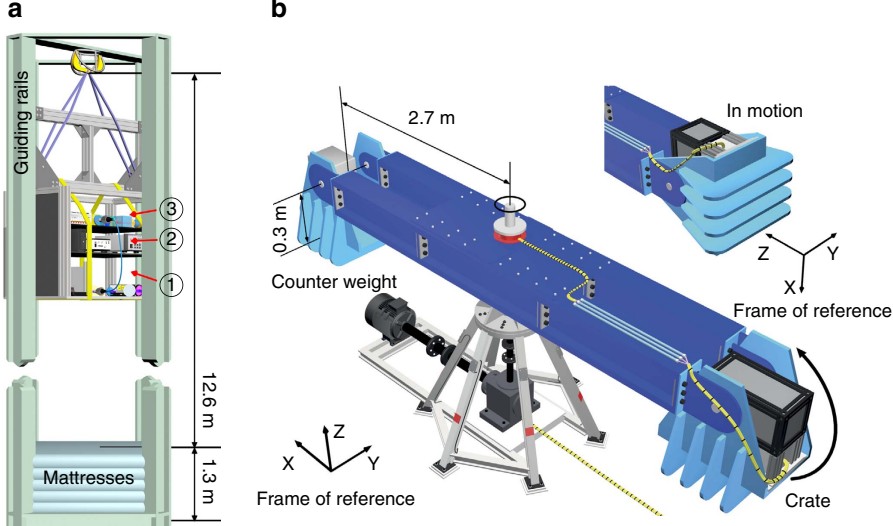

**Figure 2 | Scheme of the experiments.** (**a**) The crate was dropped from 12 m for a low-g (almost) free-fall flight in air of 1.4 s. A 1.3-m-high stack of mattresses was used to reduce deceleration on impact (Supplementary Movie 1). (**b**) The source crate was installed in one of the centrifuge gondolas at a distance of about 3 m from the axis, at full speed. The orientation of the accelerometer reference frame are shown when the centrifuge is at rest and in motion (Supplementary Movie 2).

wave plate followed by a polarizing beam splitter (PBS) is inserted in each output mode of the BS to analyse the polarization of the photons in any desired linear polarization measurement basis. Finally, the photons were detected using four passively quenched semiconductor avalanche photo diodes placed in every output mode of the two PBS. Coloured glass filters and additional interference filters were placed in front of the detectors in order to minimize background counts from the remaining pump photons and stray background light. The electronic detector signals are recorded by a time-tagging module, allowing to post-select simultaneous detection events in the two output modes of the 50/50 BS.

A lower bound on the Bell-state fidelity of the experimental state was established using the fidelity witness:

$$F_{\Psi^-}\left(\hat{\rho}_{\text{exp}}\right) = \langle \Psi^- | \hat{\rho}_{\text{exp}} | \Psi^- \rangle \geq F_{\Psi^-}^{M}\left(\hat{\rho}_{\text{exp}}\right) = \frac{1}{2}(V_{\text{HV}} + V_{\text{DA}})$$

(2)

where $V_{\text{HV}}$ and $V_{\text{DA}}$ denote the visibilities of the two-photon correlation function measured in the horizontal/vertical (diagonal/anti-diagonal) basis[16]. The two-photon polarization-correlation functions were evaluated as:

$$V = \frac{N_{13} + N_{24} - N_{14} - N_{23}}{N_{13} + N_{24} + N_{14} + N_{23}}$$

(3)

where $N$ is the number of coincidence detection events between detectors 1 and 2 for each measurement setting (HV, DA). For a pump power of $\sim 5\,\text{mW}$. The setup provides 280 kcps detected single counts in total and 7 kcps coincident counts, yielding a visibility of $V_{\text{HV}} = 97\%$ in HV basis and $V_{\text{DA}} = 96\%$ in the DA basis. Note that the visibility in the DA basis is lower, as it is affected by drifts in the alignment of polarization measurement bases, imperfect phase compensation, as well as the partial distinguishably of the two photons (for example, due to deviations from the wavelength-degenerate phase matching temperature).

**The drop tower experiment**. A drop from a height of 12 m provided us with an integration time of 1.4 s for the milli-g experiment (Fig. 2). Electrical power was provided by a battery built into the crate, and kept the source and detectors operational for about 1.5 h. A wireless network antenna provided data-connection to the crate, and was used to control the experiment. Guide rails kept the crate on track during flight and impact. Air resistance and the guide rails resulted in drag on the crate, which reached a maximum velocity of 55 km per h, enabling us to measure g-values, as low as $30 \pm 3\,\text{mg}$ after the electronic-mechanical locking mechanism was opened. To cushion the impact of the crate source, we placed a high-stack of foam mattresses at the base of the drop tower. This reduced deceleration enough to avoid permanent damage but still caused a saturation of our g-sensor at 16 g.

We observed no lasting degradation in count rates or source visibility after impact on the mattresses, and could repeat the experiment many times without the need for re-alignment between measurement runs. In each successive experimental run we measured the visibility either in the HV- or in the DA- measurement basis, as shown in Fig. 3. Each successive experimental run resulted in $\approx 400,000$ single-photon detecion events in total, which were stored as time-tags on the local computer for later evaluation. During the free-fall phase the visibility never dropped below $V_{\text{DA}} = 96\% \pm 2\%$.

**The centrifuge experiment**. For the strong acceleration regime, we used a centrifuge that rotates the crate around a fixed vertical axis, thus resulting in an outward oriented force perpendicular to the axis of rotation (Fig. 2b). The centrifuge consists of two 3-m-long arms with articulated platforms at either end (Fig. 2b). The platforms swing outwards at increasing angular velocities of the centrifuge. The crate was mounted in one of the two gondolas with a 37 kg counterweight on the other side. Electrical power and data-connection to the crate were provided by means of a sliding contact at the axis of the centrifuge to the control room.

The acceleration was varied from 1 g to up to 30 g in $\sim 5\,\text{g}$ steps by increasing the angular velocity of the centrifuge. Despite the strong acceleration the optical setup remained stable throughout the experiment. The pump beam shifted by up to two pixels on the CCD camera, which corresponds to an angular deviation of $\sim 38\,\text{arcsec}$. This is well within the alignment tolerances of the PMSM-fibre coupling lens system and no drop in the count rates was observed. For each g value (set by the angular speed of the centrifuge) we evaluated the visibility in the complementary HV and DA measurement basis for several minutes (Fig. 4). Note that a small reduction of the DA visibility stems from the fact that the temperature of the crystal was not stable at high accelerations. This was due to high-wind speed of about 174 km per h, which effectively cooled the system as indicated by the temperature sensor of the oven. The resulting deviation from the degenerate phase matching temperature introduced distinguishability of the emitted photon pair, and reduced the visibility in the DA basis accordingly.

## Discussion

We have conducted experiments with g-values corresponding to milli-g (drop tower) and hyper-g (centrifuge) settings. We used a genuine quantum mechanical system to evaluate whether the amount of entanglement is affected by non-inertial motion in accelerated reference frames. We implemented an entangled photon source together with a detection system and the required electronics in an autonomous source crate. We contiguously measured a witness, which imposes a bound on the minimum Bell-state fidelity, from the visibility of polarization correlations in two mutually unbiased basis (HV and DA).

Figure 5 summarizes the g-value versus the entanglement fidelity measured for experiments in the falling tower and the centrifuge. A Bell-state fidelity $> 96\%$ was observed for all acceleration levels. Each data point was calculated using $> 10,000$ coincidence counts leading to an $3\sigma$ error bar of 0.25%. The measured fidelity is limited by numerous sources of systematic errors such as accidental coincidence counts[17], and birefringence-induced transformations of the entangled photon state, temperature-dependent spectral characteristics of the SPDC process[18].

Our experiment can rule out any hypothetical variation of entanglement to within the precision of the current measurement apparatus. The resolution in our experiment was limited by the short experimental integration time, as well as residual drifts of the crystal temperature and alignment of the source. Note, however, that these were mainly due to wind speeds, and not correlated with the acceleration itself. The error bars shown in the graphs are calculated considering Poissonian statistics, as well as systematical errors. Due to temperature fluctuations of the nonlinear crystal ($\pm 0.1\,^{\circ}\text{C}$) and gusts of cold air in the centrifuge, an additional error of $\pm 0.6\%$ was observed in the DA-visibility. This corresponds to an uncertainty of $\pm 0.3\%$ in the Fidelity, in addition to statistical errors. The total error is well within the average fidelity taking all acquired data into account.

Note that the sources of systematic errors could be reduced by automated opto-mechanical re-alignment and better temperature stabilization.

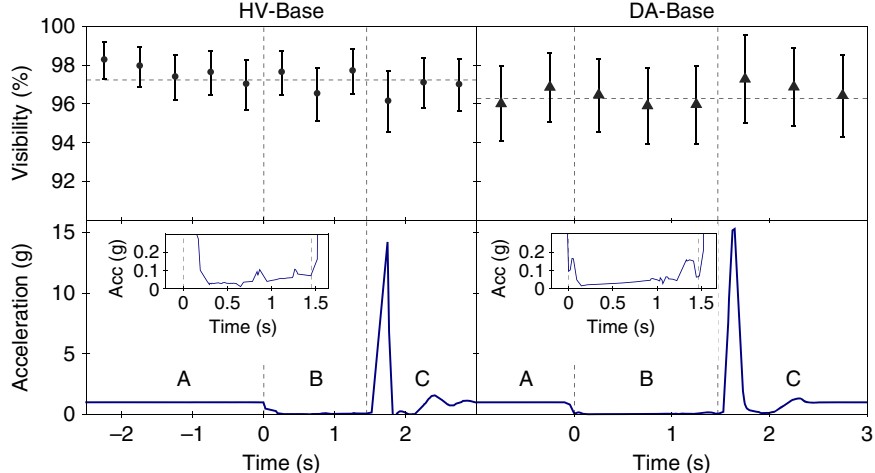

**Figure 3 | Data for drop tower experiment.** (A) Elapsed time at rest, (B) free-fall flight (C) and after impact versus HV- (left) and DA-visibility (right) measured in 500 ms slots compared to its average (denoted as horizontal dashed line). Each of data points shown consists of ≈3,500 coincident counts and $V_{DA} = 96\%$ on average. The error bars shown in the graphs are calculated considering Poissonian statistics, as well as systematical errors for DA measurements due to temperature fluctuations. The actual g-value shows the drag from contacts with the guiding rail and the wind at higher velocities. The impact (at 1.45 s) consisted of a 16 g deceleration phase (g-sensor saturated) and a rebound phase (at 2 s).

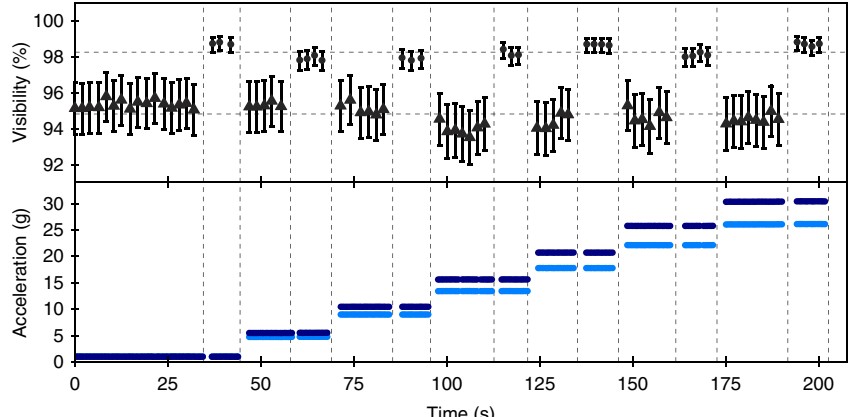

**Figure 4 | Data from centrifuge experiment.** Time elapsed versus g-force and visibility in the DA (triangles) and HV (circles) polarization measurement basis. The average of the visibilities is represented as horizontal dashed line (94.8 and 98.3%). Each of the points was calculated from ≈14,000 coincident counts. The error bars shown in the graphs are calculated considering Poissonian statistics, as well as systematical errors for DA measurements due to temperature fluctuations. The drop in visibility at 15 g is due to the effective cooling of the crystal during higher angular spinning speeds. The lower g-values (light blue) represent the measured data of the g-sensor at level 3 (detection) of the crate. In addition the calculated acceleration at level 1 (source) is plotted (blue).

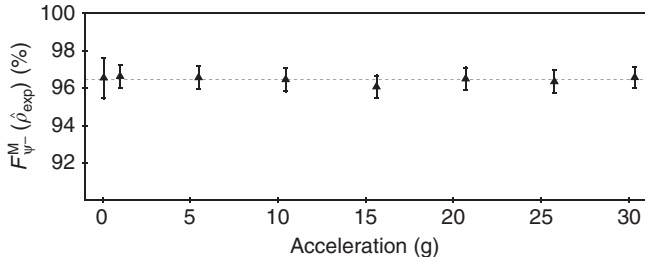

**Figure 5 | Summary of experimental data.** All data acquired during the experiments shown as the g-value versus lower bound on Bell-state fidelity $\left(F_{\psi^-}^M\left(\hat{\rho}_{exp}\right)\right)$, for g-values ranging from 3 mg up to 30 g. The error bars shown in the graphs are calculated considering Poissonian statistics, as well as systematical errors for DA measurements due to temperature fluctuations. No deviation from the total average (96.45% represented as horizontal dashed line) for more than the estimated errors is visible.

In conclusion, we have used the techniques originally developed for quantum optics high-precision experiments, to search for first experimental indications of unexpected effects in quantum systems caused by acceleration. Using the entangled system in the milli-g and hyper-g experiments described above, we found no correlation between the phase of the entangled state or its degree of entanglement with the acceleration of its non-inertial reference frame. These experiments therefore rule out any influence on the polarization-entangled two-photon quantum state that could hypothetically cause a reduction in fidelity of >1.08%.

Our study tested photonic quantum entanglement in the case of flat space-time, where the system undergoes uniform acceleration. Within this experiment we have shown that quantum entanglement should persist in a variety of accelerated frames. Apart from the inherent novelty of testing entanglement under different accelerated conditions, the results of the experiment can also be extrapolated to analogous hypothetical experiments under different gravitational fields. If, for example,

the equivalence principle holds, then a local test of entanglement should not reveal the difference between uniform acceleration and a gravitational field[19]. We thus hope that the experimental work reported in this manuscript will help in the future to shed light on quantum mechanics in non-inertial frames, ultimately including gravity.

Note that similar conditions would also accompany a rocket launch into space and are thus also relevant to subsequent quantum optics experiments carried out in space. We have shown that a full quantum optics experiment involving entangled photons was not only capable to withstand such severe stress, but also remained fully functional in severe operating conditions. Our experiment thus demonstrates the extent to which state-of-the-art quantum hardware can be exposed to such harsh operational environment, with the intention of stimulating research on theories beyond the current paradigm which can be tested with the kind of experiments presented here.

Our experimental platform represents a testbed that can readily be upgraded for measurements with higher precision, by using a ultra-bright source of entangled photons[20], and higher-dimensional degrees of freedom, such as energy-time entanglement[21]. We also envisage bringing such a system very close to zero-g conditions for several tens of seconds and reaching hyper-g up to 150 g for many hours or even day's of operation. An interesting direction for future Earth and space-based experiments is to consider photonic states undergoing non-uniform acceleration, as theoretical studies predict that changes in acceleration and changes in the gravitational field strength produce observable effects on entanglement[5–8].

**Data availability**. The data sets generated and analysed during the current study are available from the corresponding author on reasonable request.

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

## Acknowledgements

We are especially thankful to Prof. Martin Tajmar and the Technical University Dresden (Germany) for their support in the drop-tower experiment. The centrifuge experiment was supported by Jens Schiefer and Clemens Greiner from AMST in Ranshofen (Austria). We would also like to thank FFG-ALR (contract no. 844360), ESA (contract no. 4000112591/14/NL/US), FWF (P24621-N27), as well as the Austrian Academy of Sciences for their financial support.

## Author contributions

M.F. performed the experiments and analysed the data. T.S., J.H. and A.R.-A. designed the crate source. A.Z. assisted the drop tower experiment. F.S. provided experimental guidance on the entangled photon source and supervised the research. J.P., T.C.R. and I.F. provided advice and motivation regarding entanglement and the local nature of relativity. R.U. conceived the project, supervised the research and coordinated the project. All authors contributed to the writing of the paper.
