## [Peer Review File · Nature Communications]

Reviewer #1 (Remarks to the Author):

Review of manuscript entitled "Experimental test of photonic entanglement in accelerated reference frames" by Matthias Fink, Ana Rodrigues-Aramendia, Johannes Handsteiner, Abdul Ziarkash, Fabian Steinlechner, Thomas Scheidl, Ivette Fuentes, Jacques Pienaar, Tim C. Ralph, and Rupert Ursin

The manuscript reports on first systematic measurements of the impact of milli-gravity and hyper-gravity on photonic entanglement in the polarization degree of freedom. Within the error bars, the results are consistent with no effect of acceleration on the entanglement of two photons. These findings might give valuable bounds to theories on the interplay between quantum mechanics and gravity. Theories that predict an effect outside the determined confidence intervals can now be ruled out by the presented measurement results.

The measurements in the manuscript are mostly novel, with the exception of a related experiment performed on a nanosatellite in micro-gravity environment, cited as [18] in the present manuscript. Technical details, motivation and overall aims are different in the two experiments. This should be presented more in detail. The results per se are of fundamental interest to a broader audience and additionally of technical interest to specialists.

The presented work seems to be sound and interesting and therefore merits publication, after the following points are addressed:

SECTION 1:

In the first section the experiments are motivated by listing a number of theories. This part contains repetitions of similar statements. However, many references are missing here. Either one should give the relevant references or even it might be more appropriate to shorten this section by focusing on the fact that the presented work is the first systematic investigation of photonic entanglement in accelerated reference frames. The experiment could be presented as an interesting empirical result that might be used to advance theory on the interplay between quantum mechanics and gravity. In addition, statements that quantum theory and relativity theory are different with respect to the scales on which they apply, are somehow imprecise. In particular, the scaling argument is not obvious in the present experiments.

Reference [18] is cited in the context of "explosive g-forces", however, this paper describes measurements of correlated pairs on a nanosatellite in orbit. "Explosive g-forces" are described in Sci. Rep. 6, 25603 (2016) by the same group in Singapore.

SECTION 2:

In the second section the authors present the core experimental setup. The entangled photon source is described in detail and all necessary information for repeating the experiments are provided.

In line 128, the state after post-selecting events with one photon in each of the output ports of the beam splitter is lacking the phase, which is mentioned in the following section.

The formula in line 156 is wrong since post-selecting a Bell state and detecting coincidences between detector 1 and 2 are exclusive.

Fig. 1 is too small to be legible. A beam splitter itself does not split the photons randomly, but creates a superposition. The randomness comes into effect due to measuring the photons.

How distinguishable are signal and idler photons? (e.g., Hong-Ou-Mandel interference contrast or by scanning the crystal temperature).

Reduction of visibility might also stem from imperfect compensation of the relative phase due to the beam splitter.

Since stability plays a significant role: Are the polarization axes of the input beam aligned to the polarization maintaining fiber? If so, how is the alignment performed? If not, how stable is the transmission through the fiber?

SECTION 3:

In the third section, the drop tower experiment, i.e., the entangled photons subjected to a free fall, is described and the measurement results are presented. Due to drag on the crate with the experimental setup, a 30 mg environment is reached. Therefore, it might be more appropriate to use the term "milli-g" instead of "micro-g"? The results are presented in a clear and unambiguous manner.

It is not fully clear which function the data connection has. Is it used to control the experiment or to acquire measurement data during the experiment? In the latter case, why would this be preferred compared to storing data within the experimental apparatus and reading it out after the experiment?

How significant would the effect of the Coriolis force be during the short time of the free fall if no guide rails were used?

SECTION 4:

In the fourth section, the experimental setup with the polarization entangled photons is subjected to hyper-gravity equivalent accelerations. Again, the experiment is described very detailed and the results are presented in a clear way.

Fig. 2: In which way does the choice of axes have an influence on data analysis?

SECTION 5:

In the last section, the authors give a summary of the presented work and discuss the observations, including an appropriate statistical analysis. This section might also be compressed by removing repetitions. Here the authors could add a few comments on their plans to reduce systematic measurement errors, as they have done for statistical errors.

How useful could reduced-gravity aircraft be?

Are there concrete examples of theories that might directly benefit be influenced by the measurement results?

REFERENCES:

The references contain a number of typos. This should be corrected.

Is the publication status of ref. [20] known?

IN GENERAL:

English language might be checked and the authors should avoid imprecise formulations such as, e.g., "explosive g-forces".

Reviewer #2 (Remarks to the Author):

This paper reports on experiments that aim to verify entanglement for photon pairs under the influence of accelerations in the wide range from 30 mg to 30 g. It is claimed that this "work

represents the first quantum optics experiment in which entanglement is systematically tested in geodesic motion as well as in accelerated reference frames with acceleration $a \gg g$.

As such this is an interesting result, presumably worth communicating to a wider audience, given it were made clear why this is of importance to fundamental physics.

This seems to agree with the motivation by the authors, as is clearly apparent from the comparatively long introductory section "Quantum Theory vs. Relativity", the aim of which is to convince the reader that fundamental lessons can be drawn concerning the poorly understood relation between Quantum Theory (meaning either Quantum Mechanics and/or Quantum Field Theory) and Relativity (meaning General Relativity), or even Quantum Gravity proper.

I am not at all convinced by this discussion, which I think confuses many aspects of this admittedly highly interesting but at the same time also highly complex problem. Irrespectively of the correctness of the statements made in Section 1, the discussion touches upon many points which on closer inspection have nothing to do with the experiments or its interpretation. One thing it confuses is the difference between the question of how a quantum system (mechanical or field theoretic) couples to an external classical gravitational field (like that of the Earth), and the question of how a quantum system back-reacts onto itself via its own gravitational interaction (Diosi, Penrose). The experiment only relates to the former problem and has no connection to the latter. Moreover, the external classical "gravitational field" that is essential here is that produced by acceleration. Spacetime curvature is not relevant to any aspect of the experiment and the whole theoretical description could be limited to accelerated motion in flat Minkowski space. This seems to imply that what is tested here is at best the compatibility of Quantum Theory and Special Relativity. Note that non-inertial frames can of course be used in Special Relativity (like in Newtonian mechanics) and it is admittedly a perfectly legitimate question to ask how non-inertiality influences the dynamics of quantum systems. But there is still a very long way to go from here to any significant statement concerning the relation to gravity proper, i.e. Quantum (Field) Theory in classical curved spacetime, or even Quantum Gravity proper.

It is my feeling that the authors overstate the significance of their experiments as regards the theoretical problems they use as a motivation and background for it in Section 1. This mismatch takes away much of the attractiveness the paper might otherwise have for a general audience. Hence, whereas the paper may well be a suitable contribution to a specialised journal, it does in my opinion not justify to be published in Nature Communications.

Dipl. Ing. Matthias Fink
IQOQI - Vienna
Boltzmanng. 3
1090 Vienna
Austria

January 19, 2017

NCOMMS-16-19937-T, "Experimental test of photonic entanglement in accelerated reference frames"

Point-by-point reply to the reviewers' comments:

We would like to thank both Reviewers for the insightful thoughts they shared with us. We are convinced we have addressed all of the suggestions provided by both reviewers and that the manuscript has significantly improved as a consequence.

Reviewer #1:

The manuscript reports on first systematic measurements of the impact of milli-gravity and hyper-gravity on photonic entanglement in the polarization degree of freedom. Within the error bars, the results are consistent with no effect of acceleration on the entanglement of two photons. These findings might give valuable bounds to theories on the interplay between quantum mechanics and gravity. Theories that predict an effect outside the determined confidence intervals can now be ruled out by the presented measurement results.

The measurements in the manuscript are mostly novel, with the exception of a related experiment performed on a nanosatellite in micro-gravity environment, cited as [18] in the present manuscript. Technical details, motivation and overall aims are different in the two experiments. *This should be presented more in detail.* The results per se are of fundamental interest to a broader audience and additionally of technical interest to specialists.

The presented work seems to be sound and interesting and therefore merits publication, after the following points are addressed:

We thank the reviewer for this positive assessment and a number of insightful comments relating to various aspects of both technical and fundamental nature, as well as suggestions that have improved the overall presentation of the manuscript.

In response to the first comment relating to the prior experiment performed on a nanosatellite, we now state in more detail the aim and background of this

previous satellite mission. The paragraph now reads: "...carried out, where an entangled photon source was exposed to ultra high g-forces at the explosion, but not during its operation [?]. That project pioneering aimed to bring an entangled photon state on a cubesat into space, but the rocket exploded during the launch phase of the mission...."

SECTION 1: In the first section the experiments are motivated by listing a number of theories. This part contains repetitions of similar statements. However, many references are missing here. Either one should give the relevant references or even it might be more appropriate to shorten this section by focusing on the fact that the presented work is the first systematic investigation of photonic entanglement in accelerated reference frames. The experiment could be presented as an interesting empirical result that might be used to advance theory on the interplay between quantum mechanics and gravity. In addition, statements that quantum theory and relativity theory are different with respect to the scales on which they apply, are somehow imprecise. In particular, the scaling argument is not obvious in the present experiments.

The introductory section has been completely re-written and shortened in accordance with the concerns raised by the reviewer (also addressing the comments of Reviewer # 2, see below). We now make a clearer distinction between the scope of the article, which is a systematic investigation of photonic entanglement in non-inertial reference frames, and other potential experimental scenarios where new physics might be expected.

We have also added a statement to the abstract, in order to further emphasize the scope of the article: "Experimental techniques in quantum optics have only recently reached the precision and maturity required for the investigation of quantum systems under the influence of non-inertial motion, such as being held at rest in different gravitational fields, or subjected to different uniform accelerations."

Reference [18] is cited in the context of explosive g-forces, however, this paper describes measurements of correlated pairs on a nano-satellite in orbit. Explosive g-forces are described in Sci. Rep. 6, 25603 (2016) by the same group in Singapore.

The paragraph has been re-written in response to the reviewer's general comments (see below). We now also correctly cite Sci. Rep. 6, 25603 (2016).

SECTION 2: In the second section the authors present the core experimental setup. The entangled photon source is described in detail and all necessary information for repeating the experiments are provided. In line 128, the state after post-selecting events with

one photon in each of the output ports of the beam splitter is lacking the phase, which is mentioned in the following section. The formula in line 156 is wrong since post-selecting a Bell state and detecting coincidences between detector 1 and 2 are exclusive.

We fully agree with the referee's comment. We have added the beam splitter phase in the state after post-selection in line 128, and have made the formula in line 156 consistent with the detector numbering. We have also rephrased the paragraph in the main text that discusses the beam splitter: "The PMSMF guides the photon pairs to a non-polarizing beam splitter (BS) which creates an equal superposition of reflection (mode 1) and transmission (mode 2). When post-selecting events where one photon is detected in each of these two modes, one thus obtains the maximally polarization entangled state:..."

Fig. 1 is too small to be legible. A beam splitter itself does not split the photons randomly, but creates a superposition. The randomness comes into effect due to measuring the photons.

We agree and we have redistributed the elements of the figure to allow larger label sizes and improved visibility. We have re-worded the paragraph that discusses the function of the beam splitter in the caption text: "A beam splitter (BS) creates an equal superposition for reflection and transmission for each individual photon, which leads to an polarization entangled state in post selection."

How distinguishable are signal and idler photons? (e.g., Hong-Ou-Mandel interference contrast or by scanning the crystal temperature).

Polarization entanglement observed in a beam splitter source can be understood as the consequence of interference between indistinguishable polarization modes at the beam splitter. In other words, the visibility of the interference in the coherent basis ($V_{DA}=96\%$) sets a lower limit on the overlap of the signal and idler photons in all degrees of freedom (other than their polarization). The temperature of the nonlinear crystal was scanned and set to $T = 39.6^\circ C \pm 0.1^\circ C$ in order to maximize the visibility, i.e. the spectral overlap of the signal and idler photons. A typical HOM interference experiment between single photons in two distinct spatial modes was not performed.

Reduction of visibility might also stem from imperfect compensation of the relative phase due to the beam splitter.

The reviewer is correct in assuming that the relative phase between horizontal and vertical photons in the reflected arm of the beam splitter transforms the entangled state. However, this was not detrimental to the visibility since the relative phase introduced in the beam splitter was constant over the spectral bandwidth of the SPDC photons. Consequently, it could be completely

compensated for by introducing the inverse phase in the reflected arm of the beam splitter. This was achieved by angle-tuning the half-wave plate in the combination of quarter-, half-, and quarter-wave plates (QH₂Q).

Since stability plays a significant role: Are the polarization axes of the input beam aligned to the polarization maintaining fiber? If so, how is the alignment performed? If not, how stable is the transmission through the fiber?

The polarization of the photons emanating from the nonlinear crystal were aligned to the polarization axes of the polarization maintaining (PM) fiber. We have added a sentence clarifying this: "... coupled into a sole polarization maintaining single-mode optical fiber (PMSMF). The polarisation axis of the PMSMF was carefully aligned to coincide with the optical axes of the nonlinear crystal."

SECTION 3: In the third section, the drop tower experiment, i.e., the entangled photons subjected to a free fall, is described and the measurement results are presented. Due to drag on the crate with the experimental setup, a 30 mg environment is reached. Therefore, it might be more appropriate to use the term milli-g instead of micro-g? The results are presented in a clear and unambiguous manner.

We have replaced the term micro-g with milli-g throughout the manuscript.

It is not fully clear which function the data connection has. Is it used to control the experiment or to acquire measurement data during the experiment? In the latter case, why would this be preferred compared to storing data within the experimental apparatus and reading it out after the experiment?

The main purpose of the data connection was to control the experiment. We have clarified this in the corresponding paragraph: "...a wireless network antenna provided data connection to the crate and was used to control the experiment."

How significant would the effect of the Coriolis force be during the short time of the free fall if no guide rails were used?

We thank the referee for the detailed question related to the strength of the Coriolis force. The guiding rails keep the crate on track during flight and impact for a number of reasons, including: unbalance of the crate due to air resistance, release mechanism, torque at the release, ... In comparison to these effects the Coriolis force is small, and consequently we have no longer mention it explicitly in the main text.

SECTION 4: In the fourth section, the experimental setup with the polarization entangled photons is subjected to hyper-gravity equivalent accelerations. Again, the experiment is described very detailed and the results are presented in a clear way. Fig. 2: In which way does the choice of axes have an influence on data analysis?

The g-sensor was mounted next to the detectors at the swing of the centrifuge. The resulting acceleration is always in good approximation parallel to the z axis of the 3-axis accelerometer.

SECTION 5: In the last section, the authors give a summary of the presented work and discuss the observations, including an appropriate statistical analysis. This section might also be compressed by removing repetitions. Here the authors could add a few comments on their plans to reduce systematic measurement errors, as they have done for statistical errors.

We added a few points to the end of the manuscript on how we believe we can improve the errors and reliability of the experimental setting.

How useful could reduced-gravity aircraft be?

An experiment in a reduced-gravity aircraft would be intriguing, as it would allow longer data accumulation times (for each experimental run). However, since we have shown that our source can withstand repeated impacts, this could also be achieved by repeated the drop tower experiments. It is thus not immediately clear to what extent an experiment in an aircraft would improve upon our experimental configuration. Although we want to mention here, that we really want to do these kind of experiments in the future.

Are there concrete examples of theories that might directly benefit be influenced by the measurement results?

As of today, we are not aware that such theories have been developed yet. However, this was one of the main motivations for our experimental work from the beginning: to provide guidance for the development of new theories based on experimental facts. We hope our work will contribute to that in the future. We now state this more explicitly in the conclusion and thank the reviewer for making that very clear to us.

REFERENCES: The references contain a number of typos. This should be corrected. Is the publication status of ref. [20] known?

We have proof-read the references and made a number of corrections.

English language might be checked and the authors should avoid

imprecise formulations such as, e.g., "explosive g-forces".

We have re-phrased paragraphs in order to improve readability and clarity throughout the manuscript and conducted several rounds of proof reading with the help of native speaking colleagues. The term "explosive g-forces" has been removed in this process.

Reviewer #2:

This paper reports on experiments that aim to verify entanglement for photon pairs under the influence of accelerations in the wide range from 30 mg to 30 g. It is claimed that this "work represents the first quantum optics experiment in which entanglement is systematically tested in geodesic motion as well as in accelerated reference frames with acceleration $a \gg g$ ".

As such this is an interesting result, presumably worth communicating to a wider audience, given it were made clear why this is of importance to fundamental physics.

We thank the reviewer for this positive initial assessment and are glad to read that the reviewer found our results of interest for specialist as well as a wider audience. The reviewer also expressed concerns regarding the clarity with which we motivate our experimental work, and how the introductory section failed to convincingly distinguish the scope of our work from other topics in the complex problem of quantum theory and relativity. As a consequence, we have significantly revised the introductory part of the manuscript, as will be discussed in more detail in the following. We believe that the introductory now fully addresses the issues raised by the reviewer.

This seems to agree with the motivation by the authors, as is clearly apparent from the comparatively long introductory section "Quantum Theory vs. Relativity", the aim of which is to convince the reader that fundamental lessons can be drawn concerning the poorly understood relation between Quantum Theory (meaning either Quantum Mechanics and/or Quantum Field Theory) and Relativity (meaning General Relativity), or even Quantum Gravity proper. I am not at all convinced by this discussion, which I think confuses many aspects of this admittedly highly interesting but at the same time also highly complex problem. Irrespectively of the correctness of the statements made in Section 1, the discussion touches upon many points which on closer inspection have nothing to do with the experiments or its interpretation. One thing it confuses is the difference between the question of how a quantum system (mechanical or field theoretic) couples to an external classical gravitational field (like that of the Earth), and the question of how a quantum system back-reacts onto itself via its own gravitational interaction (Diosi, Penrose). The experiment only relates to the former problem and has no connection to the latter. Moreover, the external classical "gravitational field" that is essential here is that produced by acceleration. Spacetime curvature is not relevant to any aspect of the experiment and the whole theoretical description could be limited to accelerated motion in flat Minkowski space. This seems to imply that what is tested here is at best the compatibility of Quantum Theory and Special

Relativity. Note that non-inertial frames can of course be used in Special Relativity (like in Newtonian mechanics) and it is admittedly a perfectly legitimate question to ask how non-inertiality influences the dynamics of quantum systems. But there is still a very long way to go from here to any significant statement concerning the relation to gravity proper, i.e. Quantum (Field) Theory in classical curved spacetime, or even Quantum Gravity proper.

Reviewer #2 makes very legitimate complaints about our failure to clearly distinguish the experimental setting from that of e.g. gravitational back-action. In the introduction we indeed appealed to various problems that exist in putting QM and GR together as examples in order to give a broad overview of the field. The reviewer justifiably pointed out that some of these examples don't relate directly to the experimental setting - and we failed to make that sufficiently clear. Consequently, we have re-written the introduction, added new text and completely removed parts that the reviewer might have regarded as overstating the relevance of our experimental results. Specifically, our main concern is with non-inertial motion, and not directly with gravity, as we have now clearly indicated in the introductory paragraphs. We would like to thank the reviewer for pointing this out, and we believe that the changes made in response to this comment have significantly improved the manuscript.

It is my feeling that the authors overstate the significance of their experiments as regards the theoretical problems they use as a motivation and background for it in Section 1. This mismatch takes away much of the attractiveness the paper might otherwise have for a general audience. Hence, whereas the paper may well be a suitable contribution to a specialised journal, it does in my opinion not justify to be published in Nature Communications.

As stated in our response above, we addressed these important points by focusing only on the case of accelerations in a homogeneous gravitational fields, and eliminating any unwarranted references to curvature.

In order to better address these issues, we have re-written the introduction of the manuscript. We have limited the introduction to theoretical concepts closer in line with the main claims made in our experiment.

In conclusion, we thank the reviewer for the valuable insights she/he provided which we think have eliminated any confusion that might be caused to the reader.

Reviewer #1 (Remarks to the Author):

Review of revised manuscript entitled "Experimental test of photonic entanglement in accelerated reference frames" by Matthias Fink, Ana Rodrigues-Aramendia, Johannes Handsteiner, Abdul Ziarkash, Fabian Steinlechner, Thomas Scheidl, Ivette Fuentes, Jacques Pienaar, Tim C. Ralph, and Rupert Ursin

The revised manuscript has improved in several aspects, following the reviewers' comments. Some points remain to be addressed, before the manuscript can be considered for publication:

Abstract:

In the first sentence, Einstein's theory of relativity seems to refer to general relativity, not special relativity. Therefore, this introductory sentence does not seem to seem to directly relate to the topic of the paper. Reformulation of this sentence is recommended.

It is not clear which kind of precision is required for an experiment is not directly based on a quantified theory. Therefore, the statement that experimental techniques have reached the required precision might be misleading and should be reformulated.

Section 1:

There seems to be a typo in line 22 as well as in line 28 (contribute to?).

Line 50: The rocket explosion experiment seems to demonstrate very high stability of optical alignment under harsh mechanical and thermal conditions. However, it does not directly relate to properties of entanglement under these conditions since the experiment was not operational during the explosion. Therefore, this sentence does not seem to directly fit in the line of argument and should be reformulated.

Reviewer #2 (Remarks to the Author):

The authors have considered carefully the remarks and suggestions made by both referees and have brought their paper into a significantly better shape. They have also removed the confusing motivational overstatements in the introduction, which is now more truthful. The paper reports an interesting experiment in quantum optics, relating to the the question of how quantum systems behave in accelerated frames. Not much can be concluded as regards the much deeper question concerning the interaction of quantum-states of light with proper gravitational fields (space-time curvature). Because of that, I originally said in my first report that one might question why this

paper should be published in Nature, rather than in a narrower journal, like Physical Review. Modulo this reservation, the paper is now ok.

Dipl. Ing. Matthias Fink
IQOQI - Vienna
Boltzmanng. 3
1090 Vienna
Austria

February 27, 2017

NCOMMS-16-19937-T, "Experimental test of photonic entanglement in accelerated reference frames"

Point-by-point reply to the reviewers' comments: We would like to thank both Reviewers for their very constructive suggestions in both review rounds. We are also delighted to read that the reviewers conclude that our work may be suitable for publication in Nature Communications, after addressing their final comments. We are convinced we have addressed all of the suggestions provided by both reviewers and that the manuscript has improved as a consequence.

Reviewer #1:

Abstract: In the first sentence, Einsteins theory of relativity seems to refer to general relativity, not special relativity. Therefore, this introductory sentence does not seem to seem to directly relate to the topic of the paper. Reformulation of this sentence is recommended.

We have replaced "Einsteins theory of relativity" with "the theory of relativity" to avoid any confusion.

It is not clear which kind of precision is required for an experiment is not directly based on a quantified theory. Therefore, the statement that experimental techniques have reached the required precision might be misleading and should be reformulated.

Reviewer #1 is correct, the required precision remains unknown. and we have rephrased this sentence in order to reflect this.

Section 1: There seems to be a typo in line 22 as well as in line 28.

Many thanks for calling this to our attention!

Line 50: The rocket explosion experiment seems to demonstrate very high stability of optical alignment under harsh mechanical and thermal conditions. However, it does not directly relate to properties of entanglement under these conditions since the experiment was not operational during the explosion. Therefore, this sentence does not

seem to directly fit in the line of argument and should be reformulated.

The paragraph has been rearranged and the citation appears now more in the context of technological state of the art.

Reviewer #2:

The authors have considered carefully the remarks and suggestions made by both referees and have brought their paper into a significantly better shape. They have also removed the confusing motivational overstatements in the introduction, which is now more truthful. The paper reports an interesting experiment in quantum optics, relating to the the question of how quantum systems behave in accelerated frames. Not much can be concluded as regards the much deeper question concerning the interaction of quantum-states of light with proper gravitational fields (space-time curvature). Because of that, I originally said in my first report that one might question why this paper should be published in Nature, rather than in a narrower journal, like Physical Review. Modulo this reservation, the paper is now ok.

We thank the reviewer for this positive final assessment as well as the valuable feedback she/he provided, in particular, in the first review round. Without a doubt, the reviewer's input has significantly improved the manuscript. The proposed revisions helped us identify and re-write potentially confusing statements in the motivation and clearly define the scope of our experimental results. These revisions have also made our work more accessible to a broad non-specialist readership.